# Main Factors for Understanding High Impacts on CSR Dimensions in the Finance Industry

**Belen Lopez [1],\*** [iD]**, Alfonso Torres [2], Alberto Ruozzi [3] and Jose Antonio Vicente [4]**

[1] Communication Department, ESIC, Business & Marketing School, 28223 Madrid, Spain
[2] Finance and Economics Department, ESIC, Business & Marketing School, 28223 Madrid, Spain; alfonso.torres@esic.edu
[3] Business Organization and Marketing Department, Complutense University of Madrid, 28223 Madrid, Spain; aruozzi@ucm.es
[4] Market Research and Quantitative Methods Department, ESIC, Business & Marketing School, 28223 Madrid, Spain; joseantonio.vicentepascual@esic.edu
\* Correspondence: belen.lopez@esic.edu; Tel.: +34-91-452-4100

**Abstract:** The objective of this study is to explore empirically the dimensions that generate high impact in the finance industry to better understand its contribution from a Corporate Social Responsibility (CSR) perspective. We analyze data concerning impacts of finance sector firms certified by B Corp in order to identify the combinations that are necessary and/or sufficient to obtain a recognition of their high impact generation. The methodology followed to identify the impact dimensions is fsQCA, (fuzzy set Qualitative Comparative Analysis), a qualitative comparative analysis method applied to a sample of finance firms (n-181). The results indicate that financial sector firms exhibited four combinations focusing on different impact dimensions. Specifically, the first route indicates that a high degree of focus on customers and communities is sufficient to obtain a high impact score. The second path signals that the combination of the impacts on customers and corporate governance could lead to the same result, while in the third pathway the focus would be on the employees. Finally, the fourth route indicates that some financial firms focus strongly on their communities, corporate governance and their employees, but very weakly on the environmental dimension. Consequently, diverse combinations of CSR dimensions characterize financial sector contributions to impact generation and sustainable development.

**Keywords:** CSR; impacts; B-Corp; finance sector; qualitative comparative analysis (QCA)

---

## 1. Introduction

In recent years, increasing attention has been paid by academic researchers to the non-financial performance of organizations, driven by both external pressure from stakeholders [1,2] and also by firm managers initiating campaigns of awareness and action [3]. Consequently, companies have developed sustainable activities to contribute to the non-financial impact of the private sector, reporting social and environmental results in addition to their economic results. Thus, the purpose of this study is to contribute to the knowledge of some routes to positive impact in financial sector firms, potentially helping to overcome a still relatively scarce implementation of impact assessment methods [4], partially resulting from the absence of consensus in both academics' and practitioners' spheres [5].

Concerning the need to create high impact, there has been an emergence of some certifications to measure the impacts of companies, such as B Corp certification, which allow measuring impacts in different dimensions (community, governance, customers, employment and environment). In addition, they promote the generation of positive social and environmental impacts [6] of companies and, in

particular, the finance industry, related to their corporate sustainability strategies [7]. Thereby, as B Corp firms gain recognition for their high impact on their CSR dimensions, this also represents an opportunity to balance their business activities in order to pursue financial, social and environmental positive contributions [8]. Furthermore, as companies can create an impact in several ways, it is useful for academics and practitioners to analyze the different routes followed by companies in the finance sector certified as high-impact firms, showing which dimensions are relevant for firms. Moreover, to explore these dimensions, we analyze the literature concerning impacts in the finance sector, which has shown an increasing interest in CSR and sustainable development in recent years [9].

Therefore, this study aims to study the necessary and/or sufficient combinations of dimensions for the finance industry, in order for them to be recognized for their high impact generation using the B Corp dimensions. The methodology followed was fuzzy-set qualitative comparative analysis (fsQCA), and we analyzed the impact of B-Corp dimensions in a sample based on financial sector firms (n=181) in the B-Corp 2018 database, which includes the scores resulting from the certification process of these firms related to the five dimensions. This exploratory study is the first study analyzing the B Corp database with the fsQCA method, which facilitates the identification of causal relationships within the results [10]. Moreover, it provides useful information for strategic decisions of companies regarding how to afford impacts in all its dimensions and, then, communicate relevant results to its stakeholders. Furthermore, this methodology can also be useful to analyze the impacts in other industries certified by B Corp, and to better understand a company's behavior from the B Corp perspective.

The research contributes positively to the academic literature and may be useful for professionals interested in the non-financial impact of companies identifying the main dimensions, as well as the possible routes to obtain a high impact at the CSR level, attending to several stakeholders' demands [11].

This paper is structured as follows: Section 2 is a literature review on the main fields of this study, such as the concept of CSR and the strategy of companies focusing on social and environmental impacts. Thus, we analyze the concept and the main challenges of impact measurement and assessment. Next, we describe the dimensions of B Corp and the process of being recognized as a certified B Corporation.

After explaining these relevant elements of this work, we then describe in Section 3 the methodology, sample and calibration of data following the fsQCA model. Section 4 presents the results of the study, where the authors explain which B Corp dimensions are sufficient and/or necessary for high impact in finance firms. Finally, Section 5 shows the discussion and the main conclusions, including some limitations and future directions in the study of impacts.

## 2. Literature Review

### 2.1. The concept of CSR and Its Impacts

Corporate Social Responsibility has been studied by different authors and refers to the responsibility of businesses concerning society, taking into account the impacts generated by companies in their several activities [12]. The main objectives of companies are to achieve economic, social and environmental positive impacts from a CSR perspective [13] and create value for their multiple stakeholders [14], attending to the global demands of a global market. Furthermore, value creation implies benefits for both companies and society, in the current context of sustainability, in order to provide several contributions to sustainable development. The 17 Sustainable Development Goals (SDGs) described by the United Nations represent the largest initiative in the world, which involves governments, companies and institutions promoting the global objectives to achieve sustainable development. Thus, when companies integrate the SDGs in their CSR strategies, firms have positive advantages because they can create better relationships with managers and employees to improve business performance and attract business opportunities [15].

From a CSR perspective, the literature also shows how social and economic values are connected in the CSR strategies of companies [16]. Some authors have shown a positive relationship between social and economic values [17]. However, other scholars have explained that there is no empirical

relationship between both values [18–20]. Consequently, more empirical studies should clearly explain how companies can create value to their stakeholders (Freeman, 1984) based on the responsibilities of businesses to their stakeholders [21].

Furthermore, some scholars have affirmed that there exists a relationship between sustainability and CSR in companies, but with less attention to the finance sector. However, finance firms are more interested today in sustainable activities to positively influence their stakeholders [22] and, consequently, to increase the reputation of the company [23]. Thus, the combination of economic, social and environmental dimensions has a relevant role in this industry [24,25]. Moreover, other studies show how sustainable strategies in the finance sector contribute in several ways to sustainable development. In these cases, social, environmental and economic dimensions are relevant categories integrated with sustainable goals to advance social justice and social cohesion [26]. Finance firms provide several resources to obtain several impacts following the SDGs.

The concept of impact generally refers globally to the social, economic and environmental impacts of companies. Specifically, social impact is a concept also studied in the academic literature. Some authors explain that social outcomes include both social impact and social change, as well as sustaining of social benefits [27–29]. Other scholars define the concept of social impact differently, such as: "social impacts include all social and cultural consequences to human populations of any public or private actions that alter the ways in which people live, work, play, relate to one another, organize to meet their needs, and generally cope as members of society" [30]. Furthermore, social entrepreneurs use a logic model to assess their social impact. This model summarizes the relationships among inputs, processes, outputs, outcomes and impact [31]. However, there is limited knowledge about their impact from empirical studies [9,17]. Some analysis shows that even social entrepreneurship provides a tremendous opportunity to create a positive impact on society; it is necessary to measure these impacts in several ways [32]. The need for measuring impacts also includes the need for finance firms to obtain better relationships with their stakeholders [14].

## 2.2. The Concept and the Evaluation of the Impact

### 2.2.1. Conceptualization and Challenges of Impact Measurement and Assessment

There is no consensus in academic, nor in practitioner realms, as regards measurement and assessment of impacts at the theoretical and methodological levels. Regarding the terms that refer to the practice of impact assessment, the two main terms are "environmental impact assessment" (EIA), and "social impact assessment" (SIA), which appeared after the former [33].

Nonetheless, despite the explicit distinction between social and environmental impact, the term SIA has been conceived as the process by which social and or environmentally tangible benefits of a company are published, in line with the initial idea of blended value as coined by Emerson [34]. Therefore, it constitutes a methodology for defining, monitoring and communicating benefits or harms, created for specific individuals, communities and/or environments using evidence on results and social and/or environmental impacts [4]. Another related term which has been attributed a similar meaning is "social performance measurement", conceived by some authors as a comprehensive term that includes a varied set of practices (e.g., impact assessment, outcome measurement and program monitoring) that constitute the complete process of assessing social and environmental goals [35].

The above is intrinsically related to the multiplicity of methodological proposals for impact assessment and their heterogeneity in terms of the type of data they use, the kind of impacts they consider, the complexity of the model, and whether they are general or sector-specific [36]. A more general categorization of models for SIA is that proposed by Bengo and colleagues [37], which classifies them in three groups: a) those that offer a synthetic measure of impact, b) those focused on processes leading to impacts (e.g., input–output–outcome–impact) and c) those derived from the balanced scorecard.

In their recent review of the SIA field, Bonilla-Alicea and Fu found among its main challenges: a) the determination of which impacts to consider and their quantification, b) the uncertainty regarding selection, normalization and aggregation of indicators, c) the categorization of impacts, d) the consideration of global or local data and e) the stakeholders to consider [4]. These issues are problematic, because they not only do they hamper comparability and achievement of consensus, but also create a potential risk of deviation from the original social and environmental missions in order to satisfy various stakeholder needs [38]. Nonetheless, as a common characteristic, the different models associated with measurement and evaluation of different forms of impact can be considered as tools for the management of activities and communication with stakeholders, therefore constituting a means for improvement and legitimization [39].

In the present study, impact evaluation focuses on the five dimensions, which constitutes the construct of impact for B Corp: the community, corporate governance, the workers, the customers and the environment (for further detail see Table 1). Therefore, in the present study, the term impact includes the social dimension, as well as the environmental and governance ones. This multidimensional approach is in line with the current framework of well-being proposed by the OECD [40,41], which is constituted by factors related to quality of life (e.g., work-life balance, civic engagement and governance, environmental quality), material conditions (e.g., jobs and earnings) and resources for future well-being (e.g., natural, social, economic and human capital). Consequently, and from a company perspective, identifying the dimensions in which financial sector businesses have focused and leading them to obtain a certification that recognizes their high impact can increase the knowledge that characterizes the generation of positive impact by financial sector firms and allow them to communicate it to their customers through the B Corp certification. Nonetheless, to understand the interest in the above, it is important to consider the reasons for, and the relevance attributed to, the evaluation of impact.

### 2.2.2. The Relevance of Measuring and Assessing the Impact for Financial Sector Firms

Above the particularities of different activity sectors, the first main driver for evaluating impact is coping with the heterogeneous necessities of information of multiple stakeholders [37,42]. This can be explained by institutional theory [43], which posits that organizations recognize their dependence on the legitimacy, support and recognition of different agents of the external environment. According to such awareness, assessment and measurement of impact constitute a tool with the symbolic purpose of improving perceptions of stakeholders regarding the organization in order to maintain or improve reputation and access the benefits associated to it (e.g., easier access to resources and/ or higher attractiveness for customers, workers and communities).

Apart from their relevance in terms of improving organizational legitimacy, measurements and assessments of impact are also used by firms, foundations, investors, non-profit organizations and national as well as supranational institutions for organizational learning, monitorization of activities, investment decisions and/or resource allocation [44]. In summary, business is faced with an environment in which evaluation of impact is relevant for management (e.g., monitorization and improvement of performance) and for legitimization (e.g., communication of positive contributions to society and/or environment) purposes [39].

The above scenario is further complicated in the financial sector, partially because there is an emergence of a socially responsible investments market, which is paired with an increment in the number of actors concerned with non-financial impact generation through, for instance, social ventures, impact investors and intermediaries between them [45]. In terms of impact evaluation, this context is particularly heterogeneous, not only as a consequence of the inherent multidimensionality of the impact construct and of the consequent ways of generating it, but also of the absence of consensus that results from the divergence of multiple stakeholders' interests. Nonetheless, the main purpose behind assessing and measuring social, environmental and/or economic impacts generated by financial sector firms—and the one that characterizes the social finance field—is to facilitate their maximization through the increased understanding of the activities leading to them [46].

In particular, for impact investors, the aforementioned aim is paired with the recognition of the integration of social, environmental and corporate governance aspects in the risk assessment of investments, since they are conceived as signals of the efficiency of the invested company and can positively influence their investment performance [47].

As regards insurance companies and banks, although their direct impact on the environment may be small due to the particular nature of their production processes [48,49], indirect impact through the facilitation of economic activity, which in turn is partially dependent on the social and environmental risk assessments made by insurance firms, can be substantial [50]. In line with this, some authors [51] found that, in order to cope with increasing calls for adopting a triple-bottom-line (social, economic and environmental) approach [52], the impact dimensions on which insurance companies had the greatest focus were their managerial activities, followed by social and environmental activities.

In the case of commercial banks, an additional reason for evaluating impact assessment lies in increasing their attractiveness for customers that may compare them with credit cooperatives. In 2019, Aramburu and Pescador found that, for cooperative banks, the loyalty and reputation of customers in relation to CSR were explained by economic, environmental and social dimensions, whereas for commercial banks only by the latter two [53]. Furthermore, another important reason for banks to report they are making positive impacts is to restore the loss of trust in them during and after the last global financial crisis [54].

All the above point to different ways in which financial sector firms aim to increase their firm value, legitimacy and performance through measurement, assessment and communication of impacts. In addition, the multidimensionality of the impact construct is expected to be manifested in the approaches toward its generation among firms that are recognized as high impact, as suggested by a recent work [55], which found that the relationship between CSR and firm value was significant only when environmental, social and corporate governance dimensions were considered together.

### 2.3. Impact and Dimensions. B Corp Certification

Third-party audits and certifications generate opportunities for entrepreneurs in different markets [56]. Some companies certify to reduce regulatory risk, to improve efficiency or to indicate to customers their quality [57]. This provides different stakeholders with information, otherwise hidden, of the impact that they generate in the environment and society. While many certificates consider a very narrow area of business activity, B Lab audits and certifies firms employing a large variety of social and environmental indicators, following a multidimensional approach. The companies certified by B Lab are denominated Certified B Corp (or CBC), and although this initiative is still at a very early stage (it started in 2006), it may benefit society, the environment and people [58]. At the end of 2019, there were more than 3.200 CBCs, in 150 industries across 71 countries.

### 2.3.1. Companies Certified by B Corp

There are many definitions of what a Certified B Corp is in the literature. CBCs are firms that voluntarily submit to social and environmental audits conducted by B Lab [59]. These companies use market mechanisms to create public and common goods [60]. People are demanding companies that generate positive social and environmental impacts with their products [6]. These authors consider that companies have incorporated Corporate Social Responsibility (CSR) in order to satisfy the new demands of the community, creating social value and improving a firm's reputation and customer engagement. CBC is a new entrepreneurial phenomenon that operates with high standards of transparency and management to achieve good financial performance in parallel to a positive impact on society and the environment [61]. For these companies, markets are the best way to contribute to social and environmental changes, and benefits become a tool to reach their goals. Thanks to a voluntary certification process, these companies balance their mission and business activities, resulting in parallel social and environmental goals and financial benefits [8].

### 2.3.2. The B Impact Assessment and Certification Process

In order to obtain the B Corp Certificate, companies have to go through a very rigorous process-denominated B Impact Assessment (BIA). Its objective is to detect the company's strengths and weakness in five key impact areas: workers, governance, environment, customer and community (see description in Table 1). This process considers the daily company operation as well as the business model impacts.

B Lab evaluate the five impact areas based on an online form sent by the candidates to become CBCs. In case they get 80 points or more (out of 200) they can be certified. Beforehand, they will be required to modify their legal governing documents in order to balance their social, environmental and financial goals. Companies must repeat this process every three years to be certificated again.

### 2.3.3. Previous Research about Certified Benefit Corporations

Many types of research on CBCs have taken place in the last few years. It is not the intention of this document to make an exhaustive and detailed analysis of them but to demonstrate the scientific interest in CBCs' social impact and performance.

Social enterprises face difficulties reaching scale without compromising their mission. In most cases, when managers faced a dilemma between obtaining financial benefits and their social mission, they have favored the former [62].

**Table 1.** Description of impact areas according to B corp impact assessment.

| Impact Area | Description |
| --- | --- |
| Community | This dimension considers the relation between the company and its suppliers, what kind of implication it has in the community and other factors in relation to the company diversity. It evaluates if the products or services final scope helps to solve a social problem as health, basic service access or education. |
| Customer | It evaluates the impact that a firm may have on its customers. This dimension considers if the company products or services increase social welfare, satisfies population who is underserved or aim to solve environmental or social issues (e.g., protection of the environment or promotion of arts). |
| Environment | Assess the impact that the uses of the resources of a company generate in the environment (e.g., energy, inputs or facilities). This dimension includes everything related to its supply chain – including transportation or distribution channels-. It also considers if products or services have been designed to solve environmental problems (e.g., wildlife preservation, contamination improvement, reduction of waste, production of non-polluting energy or environmental education). |
| Workers | Asses the company relation with its employees. It analyzes how workers are treated by the company in various aspects, such as ownership opportunities, compensation, training or social benefits. It is very important the work environment as well as the vertical communication or the flexibility, health and security in the workplace. |
| Governance | It considers the mission of the company together with its accountability and transparency. It evaluates the environmental and social goals included in the company mission and the commitment of the stakeholders to achieve them. In this dimension, it assesses how the company shares financial information with its workers, the facilities for employees to give feedback and the diversity in the company´s governing body. |

Source: https://bcorporation.net/certification Accessed on 18 August 2019.

When CBCs treat employees (or consumers) excellently, workers productivity (or sales growth) will be higher [63]. Sales also grow more for CBCs that for their non-hybrid matched peers. As an outcome of their research, these authors state that CBC status does not reduce company profitability, but instead increases long-run value and makes it possible for communities, consumers, employees and the environment to simultaneously receive a positive impact.

CBCs pursue simultaneously social, environmental and economic outcomes through blending rather than assimilating [64]. B Corps do not seek to maximize profits, as per market logic, but profits are the means to accomplish their social purpose [65].

A significant number of studies on certification follow an external focus. They point out how certification allows firms to demonstrate their commitment to specific environmental or social problems to different stakeholders [58,66,67]. The certification may generate consumer rewards, as firms defend social causes, improving their growth opportunities. These signals must be costly, so that to receive the certificate is not possible for all the companies, but only for the authentic ones. Without this cost, the signals would be worthless, given that many of the companies certified would not have a real social and environmental commitment [67,68]. Although the monetary cost of obtaining the B Corp certificate is small (e.g. 500 € for firms with sales smaller than 150.000 €), the requirements to become a CBC are very demanding. Companies may need to dedicate a large amount of time and resources, diverting limited managerial and employee attention away from exploiting new business opportunities to grow [69,70], and instead, focusing on internal changes to satisfy B Lab requirements. This could have a negative impact on some companies' short-run growth, mainly in those which are smaller and younger, suffer from a lack of resources or have an immature organizational structure [71,72]. Other researchers have also observed a decline in the growth of the certified firms that is more noticeable in younger and smaller companies [73].

In summary, B Lab asses the social and environmental impact generated by companies through a multidimensional approach, using a wide range of environmental and social indicators classified in five dimensions (community, customers, employment, governance and workers). The review of the scientific literature in the previous sections evidences that the aforementioned dimensions are relevant to explain the impact generated by financial companies. In addition, many investigations have taken place to analyze the CBC phenomenon, although none of them has the same goal as the one considered in our paper. Based on all the previous reasons, we could reasonably argue that the B Corp approach provides a good basis to identify the different paths followed by financial sector companies that have certified their high impact generation.

## 3. Methods

### 3.1. Qualitative Comparative Analysis

The purpose of this method, designed by Ragin [74], is to establish the causal conditions and/or combinations of them (termed configurations) that lead to a specific result (termed outcome). Among the different typologies of QCA methods, the fuzzy-set variant (fsQCA) allows considering various degrees of categorization of the considered conditions. In addition, the fsQCA method is particularly well suited for the study of small and medium-size samples [75], but equally valid for bigger ones [76]. Another main feature of this method is that it considers the possibility of multiple conjunctional causality or equifinality, which is the awareness that different combinations of conditions can lead to the same outcome, which is in line with the idea that financial sector companies can generate a high impact through multiple paths.

According to Ragin [77], fsQCA allows identifying the conditions and/or combinations of them that are necessary or sufficient for a particular result. For example, if all the financial sector firms obtaining a high impact score showed a high degree of focus on the environmental dimension, then it would be a necessary condition. As regards sufficiency, this simply means that exhibiting the particular condition or combination of them is enough to achieve the result. For example, it could be that a high degree of focus on communities and corporate governance with a lower degree of it on the environment leads to a high impact score. As previously noted, impact is a construct that is constituted of and influenced by multiple aspects. Consequently, in order to determine the causes that contribute to generating impact at different levels, the qualitative comparative analysis method has been chosen,

as it has been previously used in other studies focusing on combinations of activities related to the dimensions of CSR [78].

*3.2. Sample and Calibration*

3.2.1. Selection of the Sample

An adequate sample for fsQCA must contain cases that are similar with respect to their context and outcome but heterogeneous in terms of the combinations of conditions they exhibit and their degree of presence. Consequently, the present study considered a sample of 181 financial entities pertaining to five sub-sectors as categorized by B Corp and with their certification actually in vigour: credit providers, equity investors in developing and/or developed markets, insurance companies and financial advisors.

Following a procedure recommended by some scholars [79,80], first the outcome, as well as the conditions potentially leading to it, must be specified clearly. Considering the previous literature and the BIA model, the outcome is the achievement of a high impact score among the financial sector companies with their certification actually in vigour in the B Corp database ($n = 181$), which is interpreted as a recognition of a high generation of positive impact.

In particular, the selection of the sample aimed to maximize the number of cases subject to analysis. Consequently, all certified financial sector firms with their certification actually in vigor included in the B Corp database were considered for analysis. Regarding their respective data, the last valid score for the certification was taken into account in order to use the most updated information available at the time of realization of the study (September of 2019). Particularly, the minimum score in order to be recognized as a certified B Corp is 80 points; in the sample for the present study, the scores of the included companies range between this minimum score and a maximum of 176. Furthermore, in order to determine the firms with a high or low impact score among the certified ones, we realized the calibration process, which is necessary in order to conduct the fsQCA.

3.2.2. Calibration Process

As regards the conditions leading to the outcome, they are the five dimensions of impact represented by the five B Corp pillars: the communities, the corporate governance, the workers, the customers and the environment. As regards the possible combinations between conditions, this is the result of $2^k$ where k is the number of considered conditions [77] (therefore, thirty-two ($2^5$)).

In order to analyze the cases, the degree of focus on each of the conditions shown by each firm must be determined, by categorizing them according to their impact score in each of the dimensions. This process is denominated calibration, and, in the case of fsQCA, this categorization can consider more than two groups to which cases can pertain [80].

Following Ragin [77], in the calibration done for this study (shown in Table 2), three degrees of membership were allowed: fully in, fully out, and neither in or out (also known as a point of maximum ambiguity).

**Table 2.** Calibration cut-offs.

| Impact Scores | Calibration | | | Statistics | | | |
|---|---|---|---|---|---|---|---|
| | Fully Inside | Maximum Ambiguity | Fully Outside | Max | Min | Mean | Std. Dev. |
| Overall | 128.7 | 98.1 | 82.8 | 176 | 80 | 102.8 | 19.0 |
| Community | 41.2 | 23.4 | 14.3 | 52.5 | 4.4 | 25.3 | 9.9 |
| Customers | 48.8 | 28.4 | 4.3 | 58.2 | 0 | 27.8 | 15.7 |
| Environment | 11.2 | 6.6 | 3.7 | 31 | 0,5 | 7.1 | 3.7 |
| Governance | 19.5 | 15.2 | 9.4 | 24.3 | 3.3 | 14.8 | 4.0 |
| Workers | 35.1 | 28.3 | 20.1 | 57.8 | 0 | 27.7 | 6.8 |

Note. The overall impact score is the outcome, whereas the other impact score constitutes the conditions.

As is logical, it is recommended that calibration is based on external knowledge, minimizing the use of the characteristics of the sample for the process [81]. Nonetheless, as noted by Russo and Confente [82], when data on the conditions and the outcome result from individual perceptions obtained with instruments like questionnaires, choosing the mean as the value for defining the point of maximum ambiguity is an acceptable option. As regards the threshold values for the fully in and fully out categories, the 90th and 10th percentiles were chosen, following previous work in high impact journals in the management and business fields using the fsQCA method with samples of similar size and considering multiple conditions [83,84].

## 4. Results

The last step in the fsQCA method consists in the evaluation of the quality of the obtained models and the individualized analysis of their solutions. The validity of the obtained model depends on two indicators: consistency and coverage (both covered in a range from 0 to 1).

Consistency measures the existence of a possible relationship of necessity or sufficiency between a condition, or a combination of them, and the result, considering the cases that constitute the sample [85]. For a relationship to be necessary, consistency must be equal or higher than 0.9, whereas for sufficiency relationships, the threshold is at 0.75 [77,80]. As regards coverage, it measures the degree to which a condition, or a combination of them, explains the values obtained by the financial entities considered for the analysis.

### 4.1. Analysis of Necessity

The purpose of the analysis was to find out if a focus on any of the pillars considered by B Corp is necessary for financial sector firms to obtain a high impact score, which would be interpreted as a particularly relevant factor for the achievement of a high degree of impact generation for financial sector firms. According to the values of consistency and coverage shown in Table 3, none of the conditions has a consistency score equal or higher to 0.9; neither the presence nor absence of any of them is necessary for obtaining a high impact score. This is coherent with the inherent multidimensionality of impact, which in turn leads to the possibility to generate it in multiple ways, therefore a priori reducing the necessity to focus on one of its dimensions, and allowing for a more flexible combination of focus on several of the constituents. Nonetheless, the presence of the customers dimension shows a consistency value of 0.89, which suggests that a higher degree of focus on it could be particularly relevant to the positive impact generation of financial sector firms.

**Table 3.** Analysis of necessity for the outcome "obtain a high impact score".

| Model: Obtain a High Impact Score | | |
|:---:|:---:|:---:|
| **Condition** | **Consistency** | **Coverage** |
| CUS_RD | 0.892545 | 0.692315 |
| ~CUS_RD | 0.350917 | 0.446194 |
| COM_RD | 0.683830 | 0.691763 |
| ~COM_RD | 0.565138 | 0.519831 |
| ENV_RD | 0.605963 | 0.616210 |
| ~ENV_RD | 0.613303 | 0.561469 |
| GOB_RD | 0.693807 | 0.669026 |
| ~GOB_RD | 0.551032 | 0.530529 |
| WOR_RD | 0.650573 | 0.644000 |
| ~WOR_RD | 0.572362 | 0.537187 |

Note: The "~" before a condition expresses the absence of such a condition. For the present study, it is interpreted as a lower or even absent focus on a particular dimension of impact.

### 4.2. Analysis of Sufficiency

To be deemed valid, the model must have an overall consistency higher than 0.75, which is the case (0.78). As suggested by the analysis of necessity, the analysis of sufficiency confirms that a high generation of impact among financial sector firms is achieved through the focus on at least two of its dimensions; the specific combinations that characterize the financial sector firms that obtained such scores are shown in Table 4. In particular, it should be noted that when obtaining the results of the analysis of sufficiency with fsQCA, the model allows researchers to choose among three possible types of solution: parsimonious, complex or intermediate. The main difference between them is that the intermediate solution is the only one that allows the researchers to specify if the presence or absence of a particular condition would contribute to the achievement of the outcome analyzed. Consequently for the present study, results for the intermediate solution are shown based on the assumption that the presence of all the conditions contributes to the achievement of a high impact score, since it is the result of the sum of the scores in every impact dimension; this is also coherent from a theoretical perspective, since positive impact generation is a holistic construct which results from the combination of positive impacts on different areas.

The obtained solution consists of four combinations of present or absent focus on different impact dimensions, which characterize those financial sector firms that obtain a high impact score. The most prominent dimension among the configurations is the customers, with a high degree of focus on them being present in three of the four configurations. Its relevance is further manifested in the fact that the last configuration implies a high degree of focus on the community, the workers and corporate governance, but also a little on the environment.

On the opposite side, and coherent with the particular nature of the financial sector, the impact dimension where financial sector firms exhibited the lowest focus was the environment, which appeared only in the aforementioned configuration and in the absent form, which means a lower degree of focus on it. As regards the rest of dimensions, each of them appears in two configurations.

Specifically, the first path indicates that a high degree of focus on customers and communities is sufficient to obtain a high impact score. The second signals that the combination of impacts on customers and corporate governance could lead to the same result, whereas in the third one, instead of governance, the focus would be on the employees. Finally, the fourth path indicates that some financial sector firms recognized for their high generation of impact focus strongly on communities, corporate governance and employees but very weakly on the environment.

**Table 4.** Analysis of sufficiency for the outcome "obtain a high impact score".

| Conditions | Configuration 1 | Configuration 2 | Configuration 3 | Configuration 4 |
|:---:|:---:|:---:|:---:|:---:|
| Customer | ● | ● | ● | |
| Community | ● | | | ● |
| Governance | | ● | | ● |
| Environment | | | | ○ |
| Workers | | | ● | ● |
| Consistency | 0.87 | 0.85 | 0.84 | 0.87 |
| Raw coverage | 0.59 | 0.61 | 0.57 | 0.2633 |
| Unique Coverage | 0.081 | 0.053 | 0.062 | 0.038 |
| Overall consistency | 0.774 | | | |
| Overall coverage | 0.869 | | | |

Note: the symbol "●" indicates the presence of the condition, whereas the symbol "○" indicates its absence.

In addition, in order to identify patterns within the different subsectors considered (credit providers, equity investors in developed and developing markets, insurance companies, and investment advisors), the exhibition of configurations in each case was considered in the study. The results of the exploration are as follows:

- Credit providers (*n* = 38). The 36.8% of the banks included in the B Corp database obtained a score that certified their high generation of impact, being the sector with most companies obtaining the result. Twenty-five of them showed a high degree of focus on the customers dimension and the governance or community ones, whereas the other thirteen exhibited a high degree of focus on corporate governance, communities and workers, as well as very little on the environment.
- Equity investors in developed markets (*n* = 38). In this case, 28.9% of the companies in the B Corp database obtained a high impact score, showing a pattern of configurations similar to that of credit providers.
- Equity investors in developing markets (*n* = 18). These chose to focus mainly on customers and governance, or on customers and workers, with 22.2% of them obtaining a high impact score.
- Investment advisors (*n* = 68). Of these, 26.8% exhibited high impact generation, with all of them highly focused on customers and corporate governance, or on communities or workers, with the latter being the most-chosen combination.
- Insurance companies (*n* = 17). The only one of these that obtained a high impact score exhibited a high degree of focus on workers, communities and governance, but also a very low focus on the environment.

## 5. Discussion and Conclusions

### 5.1. Discussion

In recent years, financial sector firms have experienced an increment in demands from multiple stakeholders to increase their efforts toward sustainable development and CSR activities [49,50]. This has also been coupled with increasing empirical evidence showing their positive influence on firms' performance in an indirect manner, through their positive influence on employees, which in turn improves financial performance [86]. Nonetheless, as noted by some authors, [87] the positive effects of CSR on performance can be realized after achieving a certain degree of commitment to it.

Consequently, and answering recent calls for research on CSR which jointly considers the different factors that constitute it [78], in order to contribute to knowledge regarding what are the strategies followed by financial sector firms which generate a high number of positive impacts, the main goal of

this study has been to identify the dimensions of impacts and/or combinations of them which have been the major focus of the financial sector firms that have been recognized for their high generation of impact through their B Corp certification and its associated impact score.

The dimensions considered by B Corp (community, governance, workers, customers and environment) were deemed valid proxies for answering the question, since they are in line with relevant frameworks for sustainable development [40] and cover many of the aspects considered by SDGs. In addition, the model offers a synthetic measure of impact that allows comparison between firms. Thus, obtaining B Corp certification is required to pass the audit process, therefore overcoming some of the main limitations of other methodological proposals for impact assessment [4,37].

The second goal of this research was to explore if there is any pattern regarding the focus on different dimensions among five financial sub-sectors (credit providers, equity investors in developed markets, equity investors in developing markets, investment advisors, and insurance companies). In order to achieve both goals, an fsQCA analysis was performed, of which the results are summarized in the conclusions section.

According to the literature review, more empirical studies should be done to explain how companies can create value for their stakeholders, based on the responsibilities of businesses to their stakeholders [14]. Moreover, there is limited knowledge about their impact from empirical studies [9,17]. Additionally, some analysis shows that social entrepreneurship may create a positive impact on society, but it is necessary to measure that impact in several ways [32]. The need for measuring impacts also includes the need for finance firms to obtain better relationships with their stakeholders [14].

The scenario is further complicated in the financial sector, partially because there is an emergence of a socially responsible investments market, in parallel with an increment in the number of actors concerned with non-financial impact generation through, for instance, social ventures, impact investors and intermediaries between them [45].

In terms of impact evaluation, this context is particularly heterogeneous, not only as a consequence of the inherent multidimensionality of the impact construct and of the consequent ways of generating it, but also of the resulting absence of consensus that results from the divergence of multiple stakeholders' interests. In this situation, we also remark, as have some other authors, on the convenience of further research to measure the positive effects of impact generation on stakeholders.

## 5.2. Conclusions

The main conclusions of this study are collected below. First of all, the generation of a high amount of positive impact should be understood as a complex problem, as suggested by the four combinations of focus on different impact dimensions that characterized the financial firms that obtained a high impact score, and which have been differently chosen among firms constituting each of the considered sub-sectors.

Secondly, the results suggest that it is not necessary to focus on a particular dimension of impact in order to achieve a high generation of it. Nonetheless, the focus on customers appears as a particularly relevant factor, since the only path leading to a high impact score which did not include it involved a focus on four impact dimensions, whereas in the other paths the main focus was only on two of them.

Thirdly, it should be noted that, following the customers dimension, the governance, workers and community dimensions were those in which a higher degree of focus was exhibited among the companies achieving a high impact score, with each of them appearing in two of the four configurations leading to the result.

Finally, the different pathways sufficient for achieving a high impact score reveal that the environmental dimension was the one on which financial firms had the lowest focus. Theoretically, such a result is coherent due to the particular nature of the sector, where the contributions to the environment are mainly indirect [48,49]: for example, establishing certain criteria for clients regarding environmental responsibility, or considering environmental impacts when selecting among different investment options [47]. Nonetheless, it should be noted that such indirect impacts may have been

happening without being captured, since they are a consequence of direct impacts on other dimensions like the customers or governance dimensions. A possibility for future research to further explore would be to compare the degree of focus on the impact dimension, taking into consideration not only the BIA but also other methods for impact assessment like SROI (social return on investment).

In addition, this last point highlights the particular relevance of customers as the main means through which financial sector firms generate positive impact, which could explain the aforementioned predominance of the higher degree of focus on the customers dimension among the firms obtaining a high impact score. Consequently, considering the lower degree of focus on the customers dimension (average scores for each dimension for the group of firms in the database, according to each subsector, can be seen in Appendix A), this could also explain why only one of the seventeen insurance companies considered in the sample achieved a high impact score.

As regards the limitations of the present study, first, the use of cross-sectional data does not allow establishing causality, even if the fsQCA method is specifically designed to determine complex causality [77]. Consequently, longitudinal studies that consider the influence of the timing of different dimensions of impact on the overall impact generation of firms and other relevant organizational variables are encouraged.

Secondly, given the absence of consensus at theoretical and methodological levels regarding impact assessment, the results of the present study could have been different if other models were used in the study. Therefore, comparative research between different impact assessment models could complement the results of this work. Lastly, the generalization of results regarding different subsectors should be done with caution, due to the relatively small size of each of them.

Thirdly, we consider that further discussion of theories, such as institutional and signaling theories, could also contribute to an understanding of the importance of impact dimensions in different industries. Although this article does not emphasize them, both theories may be useful to understand companies´ motivations to impact positively. Therefore, future research could consider these theories to explore the reasons and needs of companies to make a positive impact.

Considering all the above, and in order to further our understanding of the factors and strategies involved in impact generation in financial sector firms, future research could consider if there is any kind of relationship between the focuses on different B Corp dimensions and relevant variables like employee satisfaction, financial performance, and/or organizational attractiveness. In addition, it would be valuable to identify the different constituents of each of the considered dimensions in order to see if there are interaction effects between them, and explore their influence on the achievement of a high impact score.

This study contributes to the knowledge of non-financial impacts of companies, helping to overcome the scarce implementation of impact assessment methods [4]. This research also can be useful for professionals interested in identifying the main dimensions and their combinations, as well as possible paths to obtain high impact at the CSR level [11]. The results show how companies can balance their business activities to pursue positive financial, social and environmental contributions [8]. Furthermore, as firms can create impact in several ways, it is useful for academics and practitioners to analyze different routes followed by companies in the finance sector certified as high impact firms, and show which dimensions are relevant for firms. Finally, the scope of the study could be broadened by analyzing other sectors considered in the B Corp database, as well as analyzing data from a cross-cultural perspective, to identify broader patterns of social, environmental and economic impact generation across sectors and regions.

**Author Contributions:** B.L., A.T., A.R. and J.A.V. contributed similarly to the development of the research and reports. All authors have read and agreed to the published version of the manuscript.

**Funding:** This research received no external funding.

**Conflicts of Interest:** The authors declare no conflicts of interest.

## Appendix A

**Table A1.** Average scores for overall and in each dimension impact of financial sector firms.

| Sector | Overall | Comm | Cust | Environm | Govern | Work |
|---|---|---|---|---|---|---|
| Credit Provider | 110.26 | 29.87 | 28.99 | 7.46 | 16.65 | 26.90 |
| Equity Investor - Developed Markets | 100.96 | 23.21 | 26.64 | 7.62 | 14.61 | 28.62 |
| Equity Investor - Emerging Markets | 101.70 | 18.79 | 37.42 | 5.67 | 12.88 | 26.57 |
| Insurance | 86.91 | 30.03 | 2.89 | 7.87 | 16.92 | 29.16 |
| Investment Advisor | 104.62 | 24.13 | 32.62 | 6.68 | 13.67 | 27.36 |
| All financial sector | 102.79 | 25.26 | 27.80 | 7.07 | 14.77 | 27.65 |

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
