# Peer review of "Main Factors for Understanding High Impacts on CSR Dimensions in the Finance Industry"

_sustainability, doi:10.3390/su12062395_

Round 1

Reviewer 1 Report

The article presents an original analysis on BIA of B Corps within the financial sector, aimed at finding the possible combination of factors leading to high scoring in BIA.

The article is clear and well written, but in reading it some doubts emerged, that could be better addressed to further clarify the contribution of this research:

  • in the methodology section, please explain how the sample of financial firms (n.= 180) has been selected; further in the article the author(s) mention that financial firms with high score where used: please explain what is intended with "high score" (e.g. all certified firms, above 80 points in BIA or other treshold?)
  • the construction of BIA is not described, that is how the scoring of each of the five dimensions is somehow summed to arrive to the total BIA scoring? This is the main concern in my opinion. I am not an expert in fsQCA but the doubt that should be clarified is: the BIA scoring maybe depends on the different weightings that the five dimensions (customers, community etc.) have in B impact score by construct. So, is it methodologically correct to use fsQCA to explain the impact of each possible combination of dimensions or the results depends on the statistical construction of the total B impact score?
  • in analsys section (4.2.) I found sentences in lines 382-387 not clear: could the authors better explain their meaning?
  • in the conclusion section, I found a contradiction between line 464 (where environment is indicated as having a high impact) and line 469, where environment is indicated as the dimension with lower focus. I think that the low relevance of environment should be better explained by the authors: does the results could indicate that environment has lower importance due to the construction of BIA (e.g. the weight of environment is lower than that of other dimensions? is BIA mainly based on social indicators?)? or is this lower importance related to the sample of firms analysed (financial sector)?

I think that clarifying these points is necessary to understand the real contribution of this kind of analysis, and could improve the understanding of the article also for readers not expert on this specific methodology. 

Reviewer 2 Report

This is an original and interesting and valuable piece of work. I think with minor revisions it should be accepted.

My only feedback is:

  • in the opening paragraph, it is supports the content, and makes the paper even more compelling to note that there is increasing awareness not JUST of external pressure, but also firm-initiated CSR awareness and action campaigns, so I suggest: 
    • rewrite the first sentence to read: "In recent years, there has been increasing attention paid by academic researchers to the non-financial performance of organizations, driven by both external pressure of stakeholders [1, 2] and also by firm managers initiating campaigns of awareness and action [3].  
    • at the end, add the citation: Sulkowski, A. J., Edwards, M., & Freeman, R. E. (2018). Shake your stakeholder: Firms leading engagement to cocreate sustainable value. Organization & Environment31(3), 223-241.
  • customarilly a short conclusion sums-up the main points and states significance to academics and practitioners (after the discussion of results) - so please consider whether breaking-out a short conclusion section out of the discussion makes sense.
  • perhaps have an English native speaker read-through and make minor stylistic adjustments (for example, that first sentence above - in the original - was a bit awkward - there are other such examples where a sentence is okay and understandable, but could be smoother).
  • one typographical error in line 470 jumped-out it should be: "as the main means" not "as the main mean" (another example of a minor error).

Thank you for allowing me to review this. I hope you accept the changes above, polish the language a bit, and that, with these minor changes, the paper is accepted. Thanks again!

Reviewer 3 Report

The paper has potential given its area of investigation and the lack of exploration of this in the academic literature. 

My main concerns with the current paper are its readability and presentation regarding writing style and use of English language. 

The paper needs a rewrite to ensure that English is used correctly and the paper is both easy and interesting for the reader. At present given the confusing nature of some of the writing it makes the article difficult to follow and the reader can loose interest at times. 

In addition, i think a little more needs to be said around the use of the methodology as this is not entirely (in its current format) clear, measurement, data collection, analysis.

There is some broad use of the term 'stakeholder' in places which i think needs revisiting as at times i think this could be referred to more specifically as a type of stakeholder i.e. investor, lender etc. 

I am not sure if this is normal protocol for the submission to this journal but without references in text it is difficult at times to see if appropriate work has been drawn upon and i think in places this may not be the case. 

Further discussion of theory would also be a valid contribution to the rewriting of this paper as at present this is rather thin. 

My final point is there are some interesting points made in the discussion and conclusion section regarding positive influences on workforce etc however there seems to be an absence of this discussion in the literature. I think this is something would could be explored further. 

Round 2

Reviewer 3 Report

Happy that original review comments have been considered and changes made.